# LI-YOLOv8: Lightweight small target detection algorithm for remote sensing images that combines GSConv and PConv

**Pingping Yan**[1]*, **Xiangming Qi**[1], **Liang Jiang**[2]

**1** Liaoning Technical University, School of Software, Huludao, Liaoning, China, **2** Tarim University, School of Information Engineering, Alar, Xinjiang, China

☯ These authors contributed equally to this work.

\* 18698929685@163.com

## Abstract

In the domain of remote sensing image small target detection, challenges such as difficulties in extracting features of small targets, complex backgrounds that easily lead to confusion with targets, and high computational complexity with significant resource consumption are prevalent. We propose a lightweight small target detection algorithm for remote sensing images that combines GSConv and PConv, named LI-YOLOv8. Using YOLOv8n as the baseline algorithm, the activation function SiLU in the CBS at the backbone network's SPPF is replaced with ReLU, which reduces interdependencies among parameters. Then, RFAConv is embedded after the first CBS to expand the receptive field and extract more features of small targets. An efficient Multi-Scale Attention (EMA) mechanism is embedded at the terminal of C2f within the neck network to integrate more detailed information, enhancing the focus on small targets. The head network incorporates a lightweight detection head, GP-Detect, which combines GSConv and PConv to decrease the parameter count and computational demand. Integrating Inner-IoU and Wise-IoU v3 to design the Inner-Wise IoU loss function, replacing the original CIoU loss function. This approach provides the algorithm with a gain distribution strategy, focuses on anchor boxes of ordinary quality, and strengthens generalization capability. We conducted ablation and comparative experiments on the public datasets RSOD and NWPU VHR-10. Compared to YOLOv8, our approach achieved improvements of 7.6% and 2.8% in mAP@0.5, and increases of 2.1% and 1.1% in mAP@0.5:0.95. Furthermore, Parameters and GFLOPs were reduced by 10.0% and 23.2%, respectively, indicating a significant enhancement in detection accuracy along with a substantial decrease in both parameters and computational costs. Generalization experiments were conducted on the TinyPerson, LEVIR-ship, brain-tumor, and smoke_fire_1 datasets. The mAP@0.5 metric improved by 2.6%, 5.3%, 2.6%, and 2.3%, respectively, demonstrating the algorithm's robust performance.

**Data availability statement:** All relevant data are available at: https://github.com/2470589561/datasets.

**Funding:** This research was funded by the National Natural Science Foundation of China (no. 62173171).

**Competing interests:** No authors have competing interests.

## Introduction

Remote sensing imagery is extensively researched and applied across various fields, including environmental monitoring and protection, urban planning and management, and crop yield prediction. However, detecting small targets within remote sensing images presents several challenges, such as difficulties in feature extraction, complex backgrounds that can easily be confused with targets, significant deviations in predicted bounding boxes, and stringent accuracy requirements, all of which hinder precise detection. With the advancement of intelligent manufacturing in China, deep learning-based object detection methods have gained increasing prominence. Single-stage detection algorithms, exemplified by the Single Shot MultiBox Detector (SSD [1]) and You Only Look Once (YOLO [2–6]) series, have become the mainstream for small target detection in remote sensing images due to their advantages in detection speed, lower parameter counts, and high recognition rates.

In 2022, Zhang et al. [7] incorporated the Bottleneck Attention Module (BAM [8]) into YOLOv5, enhancing the focus on small target information within shallow feature maps. This approach demonstrated significant effectiveness in detecting small-scale objects but failed to control the resulting increase in the number of parameters. Similarly, Luo et al. [9] integrated an adaptive spatial feature fusion module into the neck network of YOLOv4, effectively capturing global information about small targets; however, this required substantial hardware storage capacity. In 2023, Zhao et al. [10] utilized YOLOv7 as the baseline algorithm, incorporating a small target detection head and attention mechanisms to improve detection performance for small targets on water surfaces, albeit with increased model complexity. Zhang et al. [11] developed a compact DSC-SE module that fuses deep separable convolution with SE attention, reducing the parameter volume of the insulator defect model, yet it falls short in extracting small targets. Xie et al. [12] developed a lightweight feature extraction module, CSPPartialStage, which was introduced into YOLOv7 to reduce redundant computations without compromising the accuracy of small target detection in remote sensing images; however, the computational burden remained significant. In 2024, Cheng et al. [13] introduced Omni-Dimensional Dynamic Convolution (ODConv [14]) and a global attention mechanism to suppress redundant and insignificant feature expressions; however, these techniques lacked adaptability across multiple scenarios. Finally, these methods suppress redundant and insignificant feature expressions; however, they lack adaptability across multiple scenarios. Zhu et al. [15] integrated an innovative lightweight Spatial Pyramid Dilated Convolution Cross-Stage Partial Channel (LSPHDCCSPC) module into the YOLOv7 backbone network, which bolsters the capability to extract features from small targets; however, this integration has led to a decrement in the detection and recognition accuracy of these targets.

In summary, despite significant advancements in the research and application of small target detection in remote sensing images, several challenges persist. These challenges include inadequate focus on small targets, high algorithmic complexity, increased rates of missed or false detections, and limited generalization capabilities. To address these issues, this paper proposes a lightweight small target detection algorithm for remote sensing images that integrates GSConv and PConv within the YOLOv8n framework. The main contributions of this study are as follows:

1. In the backbone network's Spatial Pyramid Pooling Fast (SPPF) module, the SiLU activation function within the CBS layer is replaced with ReLU to reduce parameter interdependencies. Additionally, RFAConv is integrated after the first CBS layer to enhance focus on sample areas, thereby improving small target recognition performance.

2. An efficient multi-scale attention mechanism (EMA) is embedded at the terminal of C2f within the neck network to capture more detailed information, enhancing the focus on the features of small target areas.

3. The original detection head is replaced with GP-Detect, a lightweight detection head designed by combining GSConv and PConv, reducing parameters and computational load.

4. The border fitting loss function of the algorithm is optimized by replacing CIoU with the Inner-Wise IoU loss function, which is designed by integrating Inner-IoU and Wise-IoU v3. This approach focuses on anchor boxes of ordinary quality through a gain allocation strategy, thereby enhancing the algorithm's generalization capability.

## Fundamentals of the YOLOv8 model

YOLOv8 represents an optimized and enhanced iteration of YOLOv5, integrating advanced technologies such as the Path Aggregation Feature Pyramid (PA-FPN) network architecture, an anchor-free design, and a decoupled head. It is available in five variants: n, s, m, l, and x, each with progressively increasing sizes and parameter counts. YOLOv8 comprises three main components: Backbone, Neck, and Head, as illustrated in Fig 1.

The backbone network comprises three modules: CBS, C2f, and SPPF [16]. The CBS module extracts initial image features, while C2f captures features at scales of 80 × 80 (S1), 40 × 40 (S2), and 20 × 20 (S3) pixels, thereby providing high-level semantic information across different scales. The SPPF module extends the receptive field to integrate multi-scale features. However, the activation function of the convolution layer in SPPF incurs high computational complexity, and the number of small target feature points captured is limited, making the model susceptible to missed detections of small targets.

The neck network comprises the Feature Pyramid Network (FPN [17]) and the Path Aggregation Network (PAN [18]). The FPN transfers deep semantic features downward, while

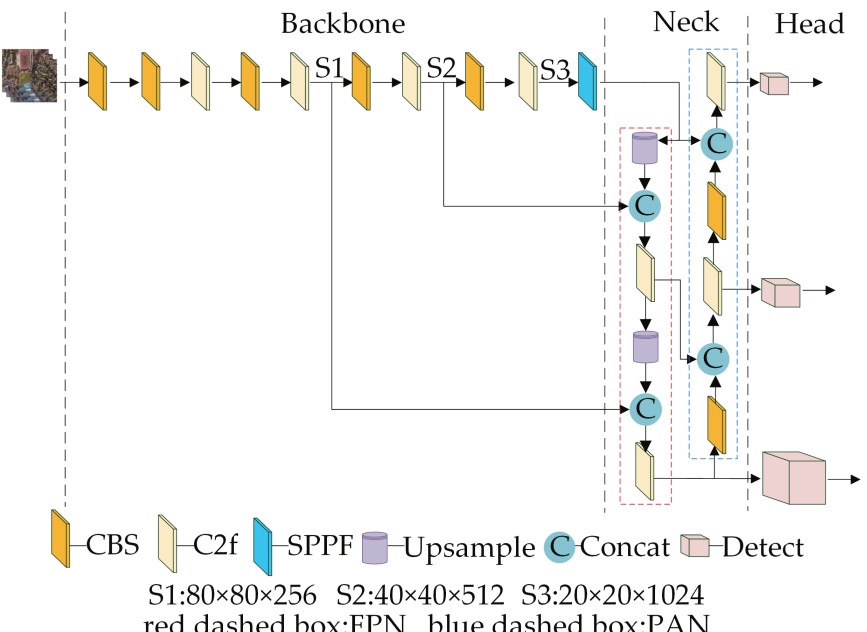

**Fig 1. YOLOv8 structure and working principle.**

the PAN propagates localization information upward. This neck network effectively integrates features across different levels, facilitating multi-scale learning that enriches the semantic information of contextual features and enhances target perception capabilities. During the feature fusion stage of the neck network, each pixel in the image undergoes multiple compressions and concatenations through CBS and Concat operations. However, during C2f feature extraction, insufficient attention to the feature areas of small targets can lead to the loss of detailed information.

The detection head primarily employs a decoupled head configuration, which segregates regression and classification tasks. Variance Focal Loss (VFL) serves as the classification loss function, balancing the weights between targets and backgrounds during the training of small target detection, thereby enhancing the predicted object class probabilities. Distribution Focal Loss (DFL), when combined with Complete Intersection over Union (CIoU [19]) as a regression loss function, rapidly focuses on regions proximal to the target to obtain accurate bounding box position information. However, the detection head contains convolutional redundancies, with its number of parameters and computational cost representing approximately 25% of the total parameters in YOLOv8n. This results in a significant computational burden when detecting small targets.

## Methods

### Algorithm implementation

The structure and working principles of LI-YOLOv8 are shown in Fig 2. SPPF-R denotes the refined Spatial Pyramid Pooling module, utilized for augmenting feature extraction of small objects. The C2f-E signifies the upgraded C2f module, aimed at elevating the focus on small objects. Lastly, GP-Detect refers to the improved detection head, engineered to decrement the network's complexity.

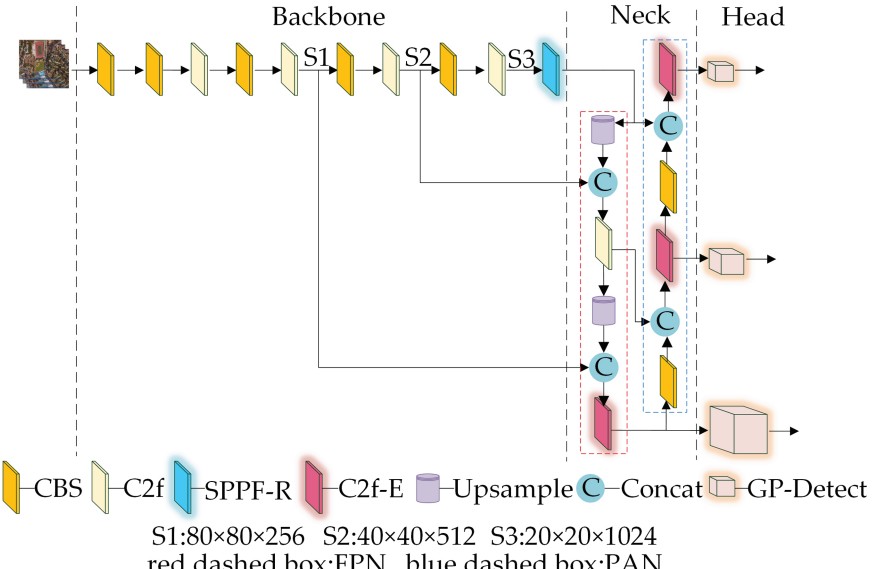

**Fig 2. Structure and working principle of LI-YOLOv8.**

## SPPF-R enhances feature extraction

SPPF processes the input feature map through CBS to capture preliminary characteristics. Given the small pixel size of targets within the receptive field area, the initial feature hbox-extraction is not comprehensive. To ameliorate this, the activation function SiLU in CBS at SPPF is replaced by ReLU, resulting in CBR. Furthermore, RFAConv is embedded in the first CBR to bolster the focus on feature information of various targets within the receptive field, thereby enhancing feature extraction. After improvement, SPPF is denoted as SPPF-R. The improvement processes for CBR and SPPF-R are shown in Figs 3 and 4.

For an input feature map of size $C \times H \times W$, RFAConv [20] first employs average pooling (AvgPool) to aggregate global features across each receptive field. It then utilizes three parallel 1×1 group convolutions (Group Conv) to rapidly extract and interact with features. This is followed by a Softmax function, which emphasizes the importance of each feature within the receptive field, thereby generating attention maps with channel dimensions of $9C \times H \times W$. Subsequently, the input feature map undergoes a 3×3 group convolution to capture spatial information within the receptive field. This process extracts and interacts with feature information, resulting in a receptive-field spatial feature map of dimensions $9C \times H \times W$. The attention map and the receptive-field spatial feature map are then reweighted and dimensionally adjusted to produce a feature map with channel dimensions of $C \times 3H \times 3W$, which is subjected to a 3×3 convolution to output an image with the same dimensions as the original feature map. The formula for RFA is expressed as:

$$F = SoftMax(g^{1\times1}(AvgPool(X))) \times ReLU(Norm(g^{k\times k}(X))) = A_{rf} \times F_{rf} \qquad (1)$$

In the formula, $g^{i\times i}$ represents a grouped convolution of size $i \times i$, denotes the size of the convolution kernel, and *Norm* represents normalization. $X$ represents the input feature map, while $F$ is obtained by multiplying the attention map $A_{rf}$ with the transformed spatial feature of the receptive field $F_{rf}$. Its working principle is expressed in Fig 5:

Therefore, the refinement process of SPPF-R is specifically analyzed as follows:

(1) SiLU replacement with ReLU. The activation function in CBS is replaced with ReLU to form CBR, ensuring that negative input values are set to zero, while positive values remain intact. The replaced activation function discards complex computations, effectively avoiding unnecessary information interference and reducing the dependency among parameters.

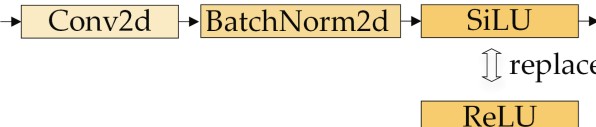

**Fig 3. Improvement of CBS to CBR process.**

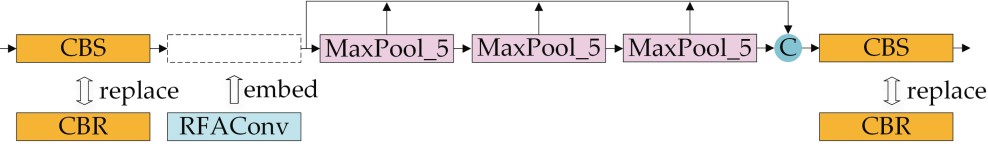

**Fig 4. Improvement of SPPF to SPPF-R process.**

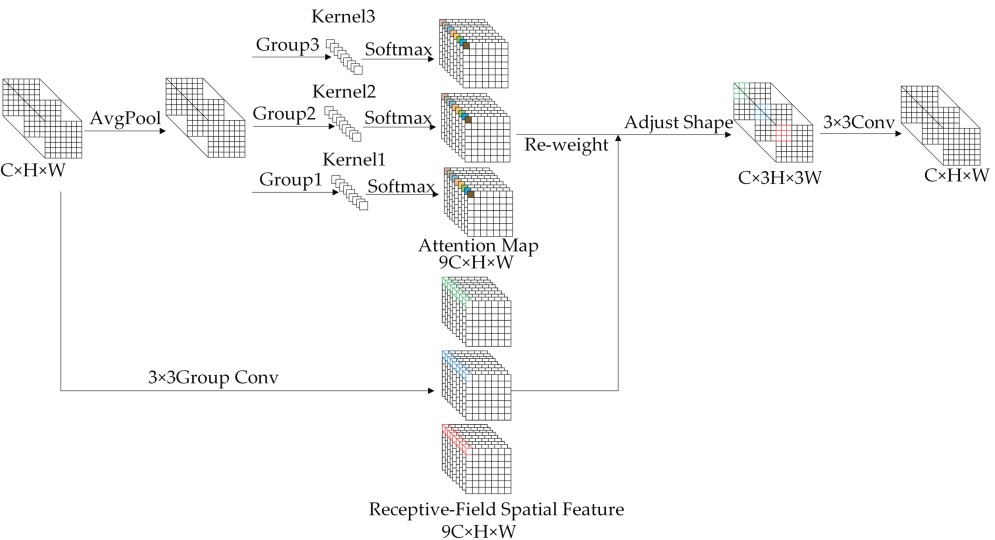

**Fig 5. RFAConv working principle.**

(2) Embedding RFAConv. The embedding of RFAConv after the first CBR integrates spatial attention mechanisms with conventional convolution to achieve flexible adjustment of convolutional kernel parameters. It also focuses on different spatial feature information within each receptive field, efficiently identifying and processing local areas in images, and markedly improving the ability to perceive and extract small targets within intricate settings.

## C2f-E enhances attention to small target areas

In the neck network, each pixel of the image undergoes multiple filtering or sliding window operations. The resulting feature-extracted images are then concatenated, which can lead to blurred feature information for small targets and the loss of critical features during deep extraction. To address these issues, the C2f module incorporates the Efficient Multi-Scale Attention (EMA) mechanism, enhancing the capture of pixel-level attention features for small targets and effectively reducing the loss of feature information. This enhanced unit is designated as C2f-E, with the improvement process depicted in Fig 6.

The Efficient Multi-Scale Attention (EMA [21]) mechanism processes an input feature map $X \in R^{C \times H \times W}$ by dividing it into G sub-features along the channel dimension, thereby facilitating the learning of diverse semantic representations. Specifically, $X = [X_0, X_i, ..., X_{G-1}]$, where $C$ denotes the number of channels, and $H$ and $W$ represent the height and width of the feature map, respectively. The working principle of EMA is illustrated in Fig 7.

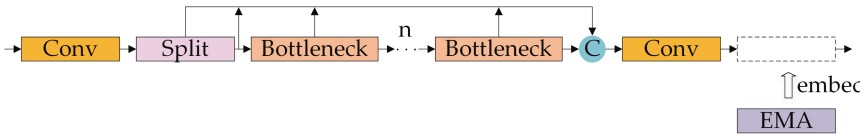

**Fig 6. C2f-E improvement process.**

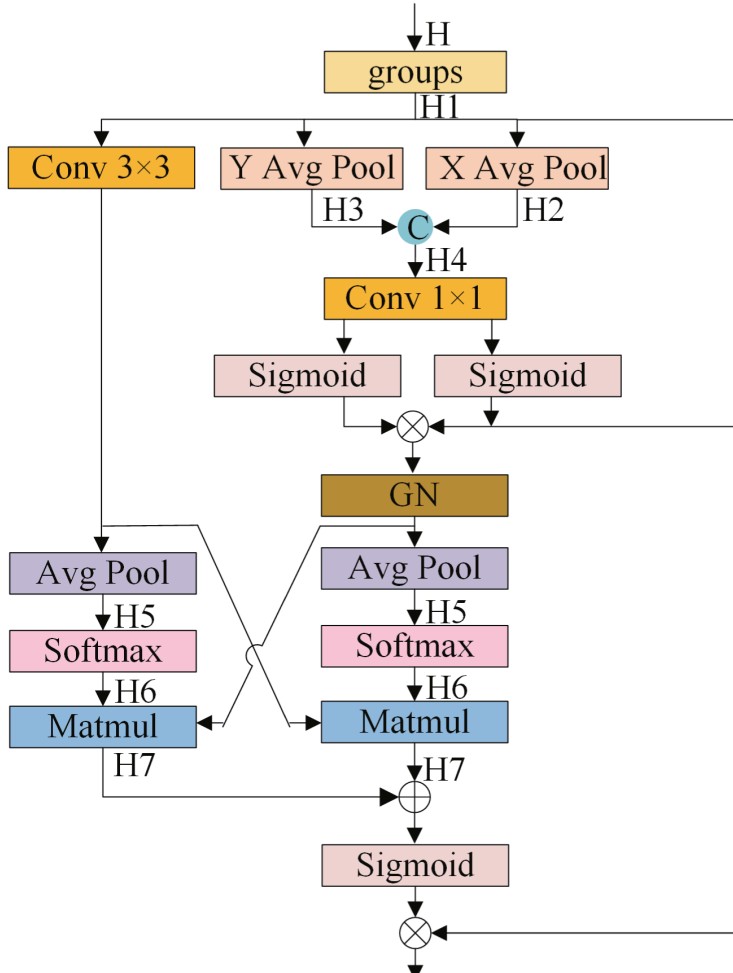

H:C×H×W  H1:C//G×H×W  H2:C//G×1×W   H3:C//G×H×1
H4:C//G×1×(H+W)  H5:C//G×1×1  H6:1×C//G  H7:1×H×W

**Fig 7. EMA attention mechanism working principle.**

EMA captures attention weights for segmented feature maps via three concurrent pathways, comprising two 1x1 and a single 3x3 branch, capturing multi-scale detail features.

First, the 1×1 branches apply 1D horizontal and 1D vertical Avg pool to encode channels along two spatial dimensions, enabling channel descriptors to accurately represent global positional information. Next, following the concatenation of spatial feature vectors along two directions, a 1x1 convolution is performed, which subsequently decomposes and outputs two distinct feature vectors. The Sigmoid operation restricts its outcomes to the interval from 0 to 1, followed by a re-weighting operation with the original feature map channel weights, which helps alleviate the imbalance between complex and simple samples among categories. The 3×3 branch captures multi-scale features using a 3×3 Conv.

After obtaining the weighted channels, GN is applied, followed by 2D global average pooling to encapsulate global spatial attributes from the outputs across each branch, producing outputs of dimensions ($R_1^{1×C//G} \times R_3^{C//G×HW}$). A SoftMax is utilized to linearly transform the

output, followed by matrix multiplication (Matmul) for local inter-channel interaction, capturing pixel-level pairwise relationships and integrating information from both directions. Finally, the values outputted from Sigmoid are reweighted with initial feature values, resulting in an output that maintains the same dimensions as the original feature map.

EMA is embedded at the end of C2f. It employs a parallel substructure to ensure that secure an uniform distribution of spatial semantic traits in each feature collective. By aggregating multi-scale spatial structural information, it mitigates the decline in small target recognition performance attributable to complex sequential processing and profound convolution. This mechanism effectively captures pixel-level attention features, establishes dependencies between dimensions, and enhances important regions within each sub-feature based on the learned weights, resulting in precise target localization information and increased attention to small target areas.

## GP-detect reduces the number of parameters and computational load

In the baseline algorithm, each of the three detection heads consists of two parallel Conv3×3 layers and one Conv1×1 layer. This configuration can result in convolutional redundancy during object localization and classification, leading to increased computational costs. By substituting the parallel Conv3×3 layers with a single GSConv3×3 and PConv3×3 structure, both the number of parameters and the computational load are significantly reduced. The enhanced detection head is termed GP-Detect, as illustrated in Fig 8.

**GSConv.** GSConv [22] downsamples a feature map with $c_1$ channels using Conv, outputting feature maps that process $c_2/2$ channels. It then employs depthwise separable convolution (DSConv [23]) to extract spatial and channel features. Integrates the feature maps resulting from the Conv and DSConv processes, followed by a shuffle operation to evenly distribute the features generated by the Conv throughout each part of the DSConv output. This results in a feature map with $c_2$ channels and its working principle is shown in Fig 9.

**PConv.** In PConv [24], feature maps with input dimensions $h \times w \times c$ are processed using filters to extract features from channels of dimension $h \times w \times c_P$, while maintaining the number of remaining channels unchanged. The processed channels are then concatenated with the unprocessed channels, resulting in an output feature map that retains the same dimensions as the original. PConv reduces redundant computations and significantly preserves the original number of channels. The working principle of PConv is illustrated in Fig 10.

In summary, GP-Detect minimizes redundant computations and reduces the number of parameters in convolutional layers, thereby enhancing the efficiency of both classification

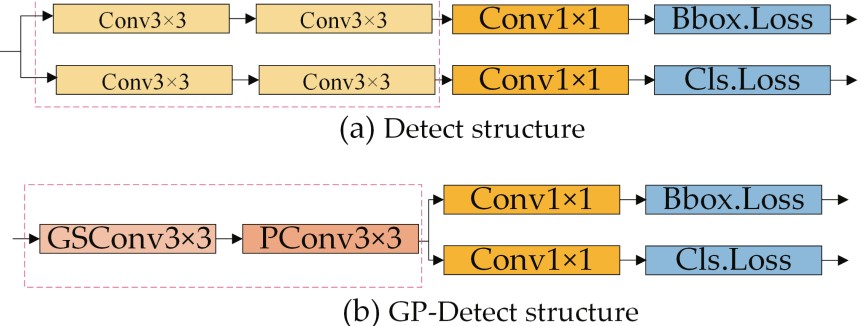

**Fig 8. Structure of Detect and GP-Detect.**

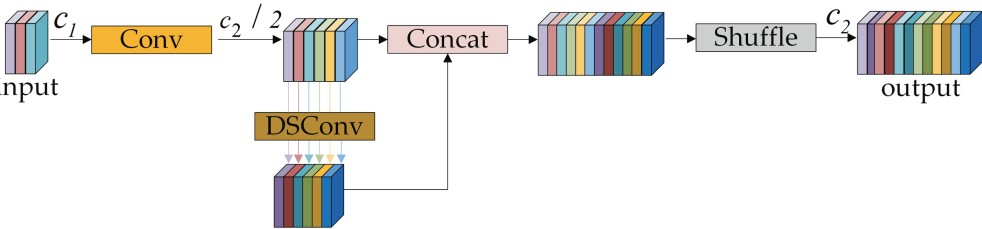

**Fig 9. GSConv working principle.**

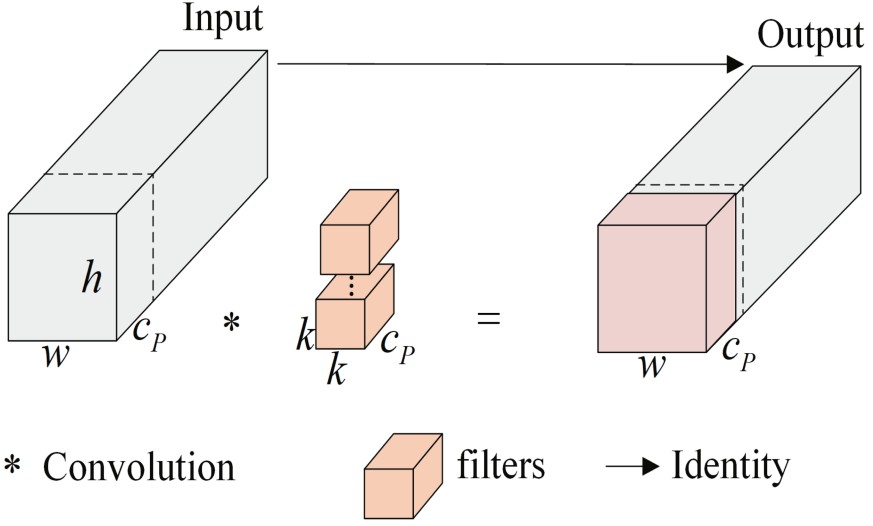

**Fig 10. PConv working principle.**

and regression tasks. The lightweight GSConv incorporates standard convolutional information into each component of depthwise separable convolutions, facilitating the effective aggregation of global information. This approach mitigates semantic information loss associated with the compression of spatial dimensions and the expansion of channel dimensions while simultaneously decreasing parameter counts and computational requirements. Additionally, PConv selectively extracts features from specific spatial dimensions, thereby lowering computational demands and parameter scales, and significantly improving the recognition of small targets.

## Inner-wise IoU enhances generalization ability

The bounding box loss function CIoU focuses solely on the distance between the centers of the ground truth and predicted boxes, along with the anchor box's width-to-height ratio. It ignores the precision of target box labeling and the equilibrium of instance distribution. Low-quality samples, under the influence of efficient fitting, affect feature learning, leading to insufficient algorithm generalization capability. Following the substitution of the CIoU with Inner-Wise IoU, the emphasis is placed on average-quality anchor boxes, thereby enhancing the algorithm's generalization capability.

**Inner-IoU.** Inner-IoU [25] generates auxiliary bounding boxes with varying sizes by controlling the scale factor ratio. It computes the overlap between these auxiliary boxes and the

ground truth boxes, facilitating more precise localization of small targets. The formula is shown as follows:

(1) Calculate the left boundary $b_l^{gt}$, right boundary $b_r^{gt}$, top boundary $b_t^{gt}$, and lower boundary $b_b^{gt}$ of the ground truth bounding box.

$$b_l^{gt} = x_c^{gt} - \frac{w^{gt} \times ratio}{2}, b_r^{gt} = x_c^{gt} + \frac{w^{gt} \times ratio}{2} \tag{2}$$

$$b_t^{gt} = y_c^{gt} - \frac{h^{gt} \times ratio}{2}, b_b^{gt} = y_c^{gt} + \frac{h^{gt} \times ratio}{2} \tag{3}$$

In the formula, $x_c^{gt}$ and $y_c^{gt}$ represent the coordinates of the center point of the ground truth bounding box, while $w^{gt}$ and $h^{gt}$ represent the width and height of the bounding box. *ratio* represents the scaling factor.

(2) Calculate the left boundary $b_l$, right boundary $b_r$, top boundary $b_t$, and lower boundary $b_b$ of the predicted bounding box (or auxiliary bounding box).

$$b_l = x_c - \frac{w \times ratio}{2}, b_r = x_c + \frac{w \times ratio}{2} \tag{4}$$

$$b_t = y_c - \frac{h \times ratio}{2}, b_b = y_c + \frac{h \times ratio}{2} \tag{5}$$

In the formula, $x_c$ and $y_c$ represent the coordinates of the center point of the predicted bounding box, while $w$ and $h$ represent the width and height of the predicted bounding box, respectively.

(3) Calculate the intersection of the auxiliary bounding box and the ground truth bounding box, denoted as *inter* .

$$inter = \left( min\left( b_r^{gt}, b_r \right) - max\left( b_l^{gt}, b_l \right) \right) \times \left( min\left( b_b^{gt}, b_b \right) - max\left( b_t^{gt}, b_t \right) \right) \tag{6}$$

(4) Calculate the union of the auxiliary bounding box and the ground truth bounding box, denoted as *union* .

$$union = \left( w^{gt} \times h^{gt} \right) * ratio^2 + (w \times h) \times ratio^2 - inter \tag{7}$$

(5) Calculate the value of the Inner loss function, denoted as $IoU^{inner}$.

**Wise-IoU v3.** Wise-IoU v3 [26] utilizes a dynamic non-monotonic focusing mechanism that can alleviate the influence of bounding box annotation quality on the generalization capability of the algorithm. The formula is shown as follows:

$$L_{Wise-IoUv3} = \frac{\beta}{\delta \alpha^{\beta-\alpha}} \exp\left( \frac{(x - x_{gt})^2 + (y - y_{gt})^2}{(w_g^2 + h_g^2)} \right) IoU \tag{8}$$

$L_{Wise-IoUv3}$ represents the loss value, exp represents the exponential function, $\delta$ and $\alpha$ are parameters, $\beta$ indicates the degree of outliers, $x$ and $y$ refer to the coordinates of the center of the predicted bounding box, while $x_{gt}$ and $y_{gt}$ represent the coordinates of the center of the

ground truth bounding box. $w_g$ and $h_g$ denote the width and height of the minimum enclosing box for both the predicted and ground truth bounding boxes. * is used for calculating separation, and $IoU$ represents the Intersection over Union (IoU) of the overlapping area between the predicted and ground truth bounding boxes.

Therefore, we design the bounding box loss function, Inner-Wise IoU, as a combination of the Inner-IoU and Wise-IoU v3 loss functions. Building on the principles of Inner-IoU, incorporating a scaling parameter to regulate the dimensions of the auxiliary bounding boxes, effectively addressing inconsistencies between bounding box size and the target shape, and generating accurate positioning information. Utilizing a dynamic non-monotonic focusing mechanism, Wise-IoU enhances the process of bounding box regression and evaluates sample quality based on the degree of outliers. As the loss value increases, this mechanism exhibits non-monotonic behavior, mitigating gradient gains for low-quality bounding boxes, while also decreasing gradient gains for high-quality anchor boxes, thus optimizing the model's learning across different quality samples. The evolution of the formula is as follows:

(1) Calculate the loss values $L_{Inner-IoU}$ and $L_{Inner-Wise-IoU}$ for the functions Inner-IoU and Inner-Wise IoU, respectively.

$$L_{Inner-IoU} = 1 - IoU^{Inner} L_{Inner-WIoU} = L_{Wise-IoUv3} + IoU - IoU^{Inner} \qquad (9)$$

(2) Calculate the degree of outlierness.

$$\beta = \frac{L_{IoU}^*}{L_{IoU}} \in [0, +\infty) \qquad (10)$$

$L^*_{IoU}$ represents the gradient gain value, and $L_{IoU}$ represents the sliding average value.

## Experiments and results analysis

### Dataset introduction

Ablation and comparative experiments were conducted on the RSOD [27] and NWPU VHR-10 [28–30] datasets, while generalization experiments were performed on the TinyPerson, LEVIR-ship, brain-tumor, and smoke_fire_1 datasets.

The RSOD was developed by Wuhan University in 2017, and intended for applications in remote sensing. The dataset is divided using a ratio of 8:2, consisting of 782 training images and 194 testing images, totaling 976 images. It encompasses four categories: aircraft, oil-tank, overpass, and playground. The size of each image varies between 512×512 pixels and 1083×923 pixels, with a total of 6,950 labeled objects.

The NWPU VHR-10 dataset, annotated by Northwestern Polytechnical University, focuses on high-resolution remote sensing. This dataset consists of 650 images of targets and 150 background images, amounting to 800 in total. From the 650 target images, 520 are selected for training and 130 for testing, following an 8:2 ratio. The dataset includes 10 categories: airplane, ship, storage tank, baseball diamond, tennis court, basketball court, ground track field, harbor, bridge, and vehicle, with a total of 3,775 target instances.

The TinyPerson dataset, published by the University of Chinese Academy of Sciences in 2019, focuses on detecting tiny individuals in distant backgrounds. It contains a total of 1,610 images, with very low resolution for the individuals, where each target has fewer than 20pixel points. The dataset includes two categories: earth_person and sea_person, with a total of 72,651 labeled instances.

LEVIR-ship is a dataset designed for the detection of small ships in medium-resolution remote sensing images. It comprises 3,896 images, including 1,973 positive samples and 1,923 negative samples. Each image has a resolution of approximately 512×512 pixels and contains a total of 3,219 annotated instances. The dataset exclusively includes the ship category.

Brain-Tumor is a dataset for brain tumor detection derived from MRI and CT scans. It comprises 893 training images and 223 validation images, totaling 1,116 images. Each image has a resolution of 512×512 pixels, and the target categories are classified as positive and negative.

Smoke_Fire_1 is a dataset developed by North China University of Technology for the detection of fire and smoke. It comprises 3,711 images, each with a resolution of 640×640 pixels. The dataset includes two categories: smoke and fire.

## Experimental environment

(1) Training Environment: NVIDIA RTX3090 GPU with 24 GB VRAM, 14 vCPUs of Intel(R) Xeon(R) Gold 6330 CPU @ 2.00 GHz, and 80 GB RAM.

(2) Testing Environment: NVIDIA RTX4060 GPU with 8 GB VRAM, 13th Gen Intel(R) Core(TM) i9-13900HX2.20 GHz, and 16 GB RAM.

(3) Software Environment: Windows 11, CUDA 11.8, PyTorch 2.0.1, and Python 3.8.0.

(4) Parameter Settings: Input feature map resolution of 640×640, all data enhanced using Mosaic data augmentation, batch size of 16, optimizer is SGD, initial learning rate of 0.01, momentum parameter set to 0.937, learning rate updated using cosine annealing learning rate schedule, training epochs set to 200.

## Evaluation metrics

The experiments utilize five frequently used assessment metrics in object recognition tasks: Precision, Recall, mAP, Parameters, and GFLOPs. The definitions and formulas for each metric are introduced below.

(1) Precision(P) represents the accuracy of recognized targets, reflecting effectiveness of the algorithm. The calculation formula is as follows:

$$Precision = \frac{TP}{TP + FP} \tag{11}$$

TP represents the quantity of true positive instances that are accurately recognized as belonging to the positive class. FP represents the quantity of false positive instances that are accurately recognized as the correct category.

(2) Recall(R) represents the recall rate of detected targets, which is the probability of predicting positive samples. FN denotes the quantity of positive instances that were not detected. The calculation formula is:

$$Recall = \frac{TP}{TP + FN} \tag{12}$$

(3) mAP represents the mean accuracy of detection across all categories. The calculation formula is:

$$mAP = \frac{1}{c} \sum_{i=1}^{c} AP_i \tag{13}$$

C represents the total number of target categories, i represents the number of detections, and AP represents the area under the PR curve for a single category. The formula for AP is:

$$AP = \int_0^1 P(R)dR \qquad (14)$$

From mAP, we derive mAP@0.5 and mAP@0.5:0.95. mAP@0.5 is the average of the average precision values (AP) at a threshold of 0.5 for all categories, while mAP@0.5:0.95 is the average mAP calculated at IoU thresholds ranging from 0.5 to 0.95 in steps of 0.05.

(4) Parameters represent the total number of parameters in the network.

(5) GFLOPs represent the count of floating-point computations, measured in G.

(6) F1-Score(F1) is the harmonic mean of precision and recall.

$$F1 = \frac{2 \times Precision \times Recall}{Precision + Recall} \qquad (15)$$

## Experimental results analysis

**Ablation experiment.** Ablation analyses were conducted using the open-access datasets RSOD and NWPU VHR-10 to evaluate LI-YOLOv8's performance. In Tables 1 and 2, I1, I2, I3, and I4 represent the algorithmic innovations SPPF-R, C2f-E, GP-Detect, and Inner-Wise IoU, respectively.

1) In the first row, the experimental results of the baseline algorithm show mAP@0.5, mAP@0.5:0.95, precision, and recall values of 84.3%, 64.9%, 90.6%, and 69.4%, respectively. The model comprises 3.0M parameters, requires 8.2G GFLOPs, and achieves an F1 of 78.6%.

2) In the second row, we introduce the innovative component SPPF-R. The incorporation of SPPF-R resulted in increases of 1.2%, 0.3%, 1.5%, 5.7%, and 4.1% in mAP@0.5, mAP@0.5:0.95, precision, recall, and F1 respectively, while maintaining the same number of parameters and computational cost. SPPF-R effectively reduces inter-parameter dependencies by replacing the original SiLU activation function with ReLU. Additionally, embedding RFAConv after the improved CBR (CBS) module expands the receptive field and enhances the extraction of small target features. The most significant improvement was observed in recall, indicating a higher proportion of correctly predicted positive samples and a reduction in missed detections caused by insufficient feature extraction.

3) the third row, we introduce the innovative component C2f-E. The incorporation of C2f-E resulted in increases of 2.5%, 0.9%, 0.4%, 2.4%, and 1.7% in mAP@0.5, mAP@0.5:0.95, precision, recall, and F1 respectively. Computational cost increased slightly from 8.2G to 8.4G, while the number of parameters remained unchanged. The mAP@0.5 metric exhibited the

**Table 1. Ablation study on the RSOD dataset.**

| N | In1 | In2 | In3 | In4 | mAP@0.5 | mAP@0.5:0.95 | P | R | Parameters | GFLOPs | F1 |
|---|-----|-----|-----|-----|---------|--------------|---|---|------------|--------|-----|
| 1 | - | - | - | - | 84.3 | 64.9 | 90.6 | 69.4 | 3.0 | 8.2 | 78.6 |
| 2 | √ | - | - | - | 85.5(↑1.2) | 65.2(↑0.3) | 92.1(↑1.5) | 75.1(↑5.7) | 3.0 | 8.2 | 82.7 |
| 3 | - | √ | - | - | 87.1(↑2.5) | 65.8(↑0.9) | 91.0(↑0.4) | 71.8(↑2.4) | 3.0 | 8.4 | 80.3 |
| 4 | - | - | √ | - | 88.8(↑1.7) | 65.0(↑0.1) | 90.9(↑0.3) | 74.0(↑4.6) | 2.7 | 6.1 | 81.5 |
| 5 | - | - | - | √ | 89.2(↑1.8) | 65.4(↑0.5) | 92.3(↑1.7) | 73.0(↑3.6) | 3.0 | 8.2 | 81.6 |
| 6 | √ | √ | - | - | 90.6(↑6.3) | 66.5(↑1.6) | 92.2(↑1.6) | 89.5(↑20.1) | 3.0 | 8.4 | 90.9 |
| 7 | √ | √ | √ | - | 91.4(↑7.1) | 66.5(↑1.6) | 95.1(↑4.5) | 86.6(↑17.2) | 2.7 | 6.3 | 90.6 |
| 8 | √ | √ | √ | √ | 91.9(↑7.6) | 67.0(↑2.1) | 95.1(↑4.5) | 86.0(↑16.6) | 2.7 | 6.3 | 90.3 |

most significant improvement. By embedding the EMA attention mechanism within C2f, the model effectively captures multi-scale spatial structural information, accurately locates small target regions, reduces feature information loss, and enhances both the detection rate and accuracy of small targets.

4) In the fourth row, we introduce the innovative component GP-Detect. The computational cost was reduced by 2.1 GFLOPs, while mAP@0.5, mAP@0.5:0.95, precision, recall, and F1 increased by 1.7%, 0.1%, 0.3%, 4.6%, and 2.9%, respectively. The number of parameters decreased to 2.7M. GP-Detect reduces redundant computations by pruning convolutional layers and enhances feature extraction capabilities by combining GSConv and PConv, significantly improving the detection rate of small targets.

5) In the fifth row, we introduce the innovative component Inner-Wise IoU. Without altering the number of parameters or computational cost, Inner-Wise IoU led to improvements of 1.8%, 0.5%, 1.7%, 3.6%, and 3.0% in mAP@0.5, mAP@0.5:0.95, precision, recall, and F1, respectively. By replacing the original loss function with Inner-Wise IoU, the loss is computed using auxiliary bounding boxes at different scales, and an intelligent gradient gain distribution strategy is employed, enhancing the algorithm's generalization capability. Maintaining the network's complexity, this approach better balances precision and recall, thereby improving the F1. It not only accurately identifies more positive samples but also achieves higher accuracy in recognizing these positive samples.

6) In the sixth row, we introduce the innovative components SPPF-R and C2f-E. The incorporation of SPPF-R and C2f-E resulted in improvements of 6.3%, 1.6%, 1.6%, 20.1%, and 12.3% in mAP@0.5, mAP@0.5:0.95, precision, recall, and F1, respectively, while maintaining the number of parameters unchanged and increasing computational cost by 0.2G. SPPF-R enhances feature extraction capabilities, thereby improving initial detection accuracy, whereas C2f-E optimizes feature fusion to further increase the focus on small target regions. The integration of these two components significantly enhances Recall, ensuring more accurate detection of targets.

7) In the seventh row, we integrate the innovative components SPPF-R, C2f-E, and GP-Detect to further streamline the algorithm while enhancing the extraction and focus on small target features. This integration resulted in improvements of 7.1%, 1.6%, 4.5%, 17.2%, and 12.0% in mAP@0.5, mAP@0.5:0.95, precision, recall, and F1, respectively, while reducing the number of parameters and computational cost to 2.7M and 6.3G. By further incorporating GP-Detect, the algorithm maintains high precision while decreasing both parameter count and computational load, thereby rendering the overall algorithm more lightweight.

8) In the eighth row, we integrated all the aforementioned innovations, resulting in significant enhancements to the algorithm's performance. The accuracy metrics mAP@0.5 and mAP@0.5:0.95 increased by 7.6% and 2.1%, respectively, while Precision, Recall, and F1 scores improved by 4.5%, 16.6%, and 11.7%, respectively. The number of parameters was slightly reduced from 3.0M to 2.7M, and the computational cost decreased substantially from 8.2G to 6.3G, marking a reduction of 1.9G (23.2%). These results demonstrate that the proposed algorithm significantly enhances the accuracy of small object detection and markedly reduces network complexity.

(2) Ablation experiments on the NWPU VHR-10 dataset

Similarly, as the four innovations contribute uniformly to the algorithm, the ablation experiments conducted on the RSOD dataset are thoroughly detailed. Experiments carried out on different datasets yield varying results. To validate the effectiveness of these innovations, eight experiments were performed using different datasets. The experimental results are presented in Table 2, where rows marked with "√" denote the innovations and their corresponding evaluation metrics. The number of parameters and computational costs of the

**Table 2. Ablation study on the NWPU VHR-10 dataset.**

| N | In1 | In2 | In3 | D4 | mAP@0.5 | mAP@0.5:0.95 | P | R | F1 |
|---|-----|-----|-----|-----|---------|--------------|---|---|-----|
| 1 | - | - | - | - | 84.3 | 53.8 | 92.0 | 74.6 | 82.4 |
| 2 | √ | - | - | - | 84.8(↑0.5) | 53.9(↑0.1) | 93.0(↑1.0) | 75.1(↑0.5) | 83.1 |
| 3 | - | √ | - | - | 85.7(↑0.7) | 54.1(↑0.3) | 93.3(↑1.3) | 75.1(↑0.5) | 83.2 |
| 4 | - | - | √ | - | 84.6(↑0.3)) | 53.9(↑0.1) | 93.0(↑1.0) | 74.8(↑0.2) | 82.9 |
| 5 | - | - | - | √ | 85.3(↑1.0) | 54.0(↑0.2) | 92.1(↑0.1) | 75.3(↑0.7) | 82.9 |
| 6 | √ | √ | - | - | 85.8(↑1.5) | 55.3(↑1.5) | 93.8(↑1.8) | 75.7(↑1.1) | 83.8 |
| 7 | √ | √ | √ | - | 86.7(↑2.4) | 54.8(↑1.0) | 95.3(↑3.3) | 75.3(↑0.7) | 84.0 |
| 8 | √ | √ | √ | √ | 87.1(↑2.8) | 54.9(↑1.1) | 95.8(↑3.8) | 76.9(↑2.3) | 85.3 |

algorithm remain consistent with those in Table 1 and are therefore omitted. The analysis is as follows.

1) In the first row, the baseline algorithm achieved experimental results with mAP@0.5, mAP@0.5:0.95, precision, recall, and F1 of 84.3%, 53.8%, 92.0%, 74.6%, and 82.4%, respectively.

2) In the second row, incorporating the innovation SPPF-R resulted in improvements of 0.5%, 0.1%, 1.0%, 0.5%, and 0.7% in mAP@0.5, mAP@0.5:0.95, precision, recall, and F1, respectively. The most significant improvement was observed in the F1 score, indicating a reduction in false positives while maintaining a high recall rate.

3) In the third row, the addition of the innovation C2f-E led to enhancements of 0.7%, 0.3 1.3%, 0.5%, and 0.8% in mAP@0.5, mAP@0.5:0.95, precision, recall, and F1, respectively. Precision was notably increased, further enhancing the accuracy of small object detection.

4) In the fourth row, the incorporation of GP-Detect resulted in increases of 0.3%, 0.1%, 1.0%, 0.2%, and 0.5% in mAP@0.5, mAP@0.5:0.95, precision, recall, and F1, respectively. Precision saw the highest improvement, thereby reducing false positive rates.

5) In the fifth row, integrating Inner-Wise IoU led to significant enhancements of 1.0%, 0.2%, 0.1%, 0.7%, and 0.5% in mAP@0.5, mAP@0.5:0.95, precision, recall, and F1, respectively. The most notable improvement was in mAP@0.5, increasing the detection rate of small objects.

6) In the sixth row, simultaneous inclusion of SPPF-R and C2f-E innovations resulted in substantial improvements of 1.5%, 1.5%, 1.8%, 1.1%, and 1.4% in mAP@0.5, mAP@0.5:0.95, precision, recall, and F1, respectively. All evaluation metrics demonstrated significant enhancements.

7) In the seventh row, combining SPPF-R, C2f-E, and GP-Detect innovations led to increases of 2.4%, 1.0%, 3.3%, 0.7%, and 1.6% in mAP@0.5, mAP@0.5:0.95, precision, recall, and F1, respectively. While mAP@0.5:0.95 and recall saw slight improvements, mAP@0.5 and precision showed more pronounced increases, enhancing the reliability and accuracy of the algorithm.

8) In the eighth row, integrating all aforementioned innovations resulted in improvements of 2.8% and 1.1% in mAP@0.5 and mAP@0.5:0.95, respectively, and 3.8%, 2.3%, and 2.9% in precision, recall, and F1, respectively. These results further confirm that the proposed algorithm significantly enhances the detection rate of small objects in object detection tasks.

**Comparison experiment.** (1) PR Curve Comparison

1) To better demonstrate the effectiveness of the LI-YOLOv8 algorithm, we conducted training on the RSOD dataset. The outcomes are detailed within Fig 11, followed by analysis.

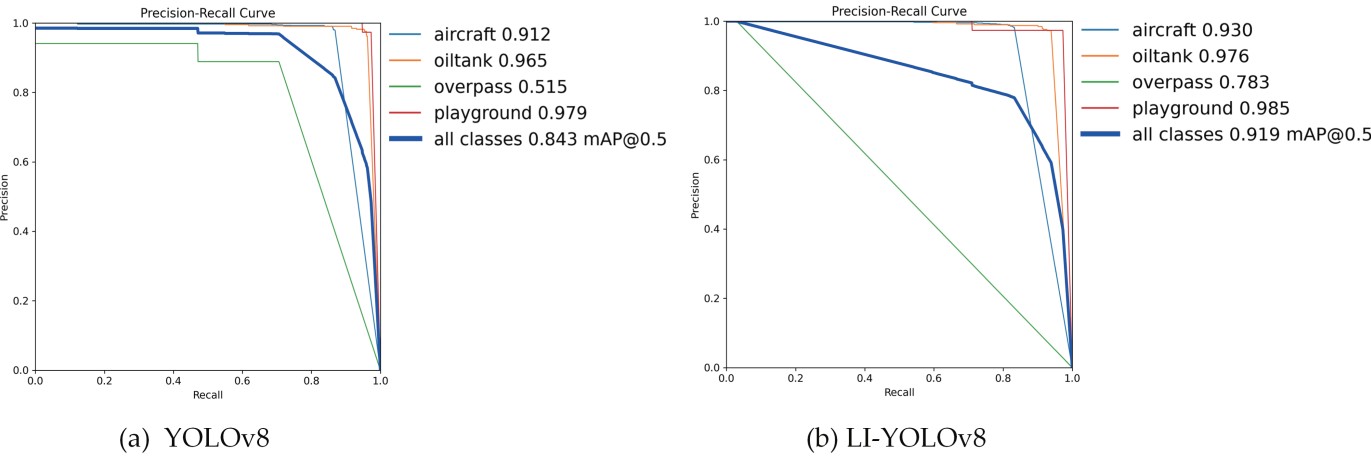

(a) YOLOv8 (b) LI-YOLOv8

**Fig 11. PR comparison on RSOD dataset.**

Fig 11(a) and 11(b) present a comparison of the Precision-Recall (PR) curves obtained from training YOLOv8 and LI-YOLOv8 on the RSOD dataset. As shown in Fig 11(a), when Recall ranges from 0.2 to 0.6, Precision remains at a relatively high level. However, when Recall increases to between 0.7 and 0.8, the Precision curve exhibits a noticeable decline. This indicates that as YOLOv8 attempts to recall more positive samples, it becomes more susceptible to misclassifying background or similar interfering information, leading to an increase in false positive rates, especially when detecting small objects in complex backgrounds such as overpasses. Additionally, the latter part of the curve features a relatively stable interval, suggesting that YOLOv8 maintains strong detection performance for certain targets (e.g., oil tanks) that occupy a significant portion of the image.In contrast, Fig 11(b) displays the PR curve for LI-YOLOv8 under the same dataset and training conditions, highlighting two significant differences:

Firstly, When Recall is between 0.7 and 0.8, the Precision curve of LI-YOLOv8 remains high without a noticeable decline, unlike YOLOv8. This suggests that LI-YOLOv8 can effectively distinguish between positive samples and the background even when recalling more targets, demonstrating enhanced small object detection capabilities in complex scenarios with fewer false positives due to the integration of innovative modules.

Secondly, When Recall ranges from 0.8 to 1.0, LI-YOLOv8 experiences only a slight decrease in Precision compared to YOLOv8, with a smaller magnitude of decline. This implies that LI-YOLOv8 continues to perform effectively in small object detection without significant increases in false negatives or false positives, highlighting the effectiveness of feature enhancement and attention reinforcement specifically targeted at small objects.

Overall, the differences between Fig 11(a) and 11(b) indicate that by integrating modules such as SPPF-R, C2f-E, GP-Detect, and Inner-Wise IoU, LI-YOLOv8 achieves a better balance between Precision and Recall over a broader range. This enhancement leads to improved overall detection performance and demonstrates LI-YOLOv8's superior adaptability to multi-scenario and multi-scale small object detection tasks.

Table 3 shows the AP improvement rates for four target categories: aircraft, oiltank, overpass, and playground. All four categories demonstrate increased AP values, with overpass showing a significant improvement of 26.8%. Aircraft and oiltank have modest increases of

**Table 3. AP improvement rates for various targets on the RSOD dataset.**

| Category | YOLOv8 AP (%) | LI-YOLOv8 AP (%) | AP improvement rate (%) |
|---|---|---|---|
| aircraft | 91.2 | 93.0 | 1.8 |
| oiltank | 96.5 | 97.6 | 1.1 |
| overpass | 51.5 | 78.3 | 26.8 |
| playground | 97.9 | 98.5 | 0.6 |

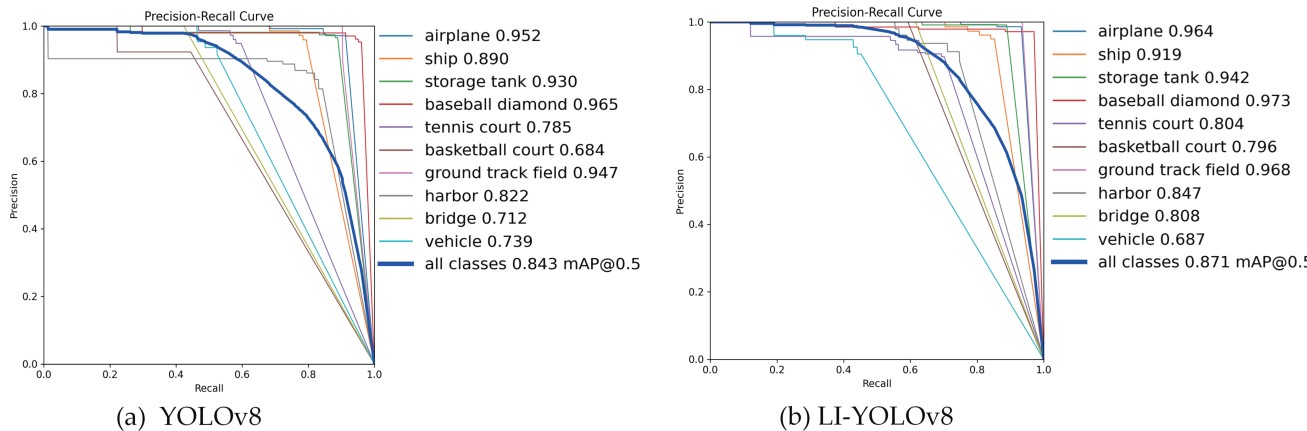

(a) YOLOv8 (b) LI-YOLOv8

**Fig 12. PR comparison on NWPU VHR-10 dataset.**

1.8% and 1.1%, respectively. The playground category, however, exhibits a minimal increase of only 0.6%, due to the baseline AP values being relatively high in the dataset, which resulted in saturation of learning capacity during training. Overall findings demonstrate that LI-YOLOv8 excels in detecting small objects in remote sensing imagery.

2) The training results on the NWPU VHR-10 dataset are shown in Fig 12, followed by analysis.

Fig 12(a) and 12(b) display the Precision-Recall (PR) curves for YOLOv8 and LI-YOLOv8 trained on the NWPU VHR-10 dataset. Overall, the trends observed are similar to those in Fig 11(a); however, Precision decreases more markedly when Recall exceeds 0.6. This phenomenon is attributable to the NWPU VHR-10 dataset encompassing a larger number of multi-scale targets (such as harbors and vehicles) and possessing higher background complexity compared to the RSOD dataset. For YOLOv8, maintaining Precision at higher Recall levels necessitates greater compromises, often resulting in a rapid decline in Precision. In contrast, Fig 12(b) illustrates the training results of LI-YOLOv8 on the same dataset, where the PR curve demonstrates a more stable and higher Precision across the board, primarily in two aspects:

Firstly, When Recall surpasses 0.6, Precision no longer experiences a significant decline but remains relatively stable. This indicates that LI-YOLOv8 possesses enhanced adaptability to small objects and complex backgrounds, thereby mitigating the risk of false positives associated with high Recall.

Secondly,When Recall exceeds 0.8, Precision continues to be maintained at a relatively substantial level. This suggests that LI-YOLOv8 effectively distinguishes similar backgrounds even when attempting to capture more targets, thereby significantly reducing both false positives and false negatives.

These differences further indicate that the enhancements made to the algorithm structure and loss function enable LI-YOLOv8 to sustain higher Precision under high Recall conditions. In summary, Fig 12(a) and 12(b) unequivocally demonstrate the adaptability and robustness of LI-YOLOv8, achieving a superior balance between maximizing Recall and maintaining high Precision in the detection of small targets within remote sensing images.

Table 4 displays the AP improvement rates for ten types of targets, including airplane, ship, and storage tank. The table indicates that the AP values for nine target categories have increased. The categories of airplane, ship, storage tank, baseball diamond, tennis court, basketball court, ground track field, harbor, and bridge saw significant improvements, while the vehicle category, due to its small proportion in the images and poor learning ability during the training process, has decreased by 5.2%. Overall, the results demonstrate that the LI-YOLOv8 performs exceptionally well in detecting small objects in remote-sensing photographs.

(2) Visual Comparison

1) On the RSOD dataset, the results comparing the LI-YOLOv8 algorithm's performance with that of the YOLOv8 algorithm are shown in Fig 13, and the analysis is as follows.

In complex background scenarios, YOLOv8 mistakenly identifies white smoke in the air as oiltank, whereas LI-YOLOv8 does not does not misclassify white smoke as a fuel tank. In scenes with tiny targets, YOLOv8 exhibits a high number of missed detections for tiny oiltank in the image, while the LI-YOLOv8 is capable of detecting a greater number of these tiny targets. For multi-scale targets, YOLOv8 does not detect the small playground on the right side, while LI-YOLOv8 can more accurately detect these small-scale targets.

2) On the NWPU VHR-10 dataset, the contrast of detection performance between LI-YOLOv8 and YOLOv8 is shown in Fig 14, and the analysis is provided below.

In complex background scenes, YOLOv8 exhibits extensive missed detections of the small target tennis court and mistakenly classifies a blue warehouse as a tennis court. In comparison, LI-YOLOv8 not only detects more small targets in complex backgrounds but also accurately identifies small targets within tennis courts. In scenes with tiny targets, LI-YOLOv8 detects more low-pixel tiny ships compared to YOLOv8. For multi-scale targets, YOLOv8 failed to detect the tennis court located above the playground, yet LI-YOLOv8 was able to identify a greater number of additional small targets.

3) Heatmap Comparison

Fig 15(a), 15(b), and 15(c) represent the original image, the heatmap of YOLOv8, and the heatmap of LI-YOLOv8. The color gradient varies from blue to red, reflecting the model's attention to targets from low to high. In comparison with the baseline algorithm, LI-YOLOv8

**Table 4. AP improvement rates for various targets on the NWPU VHR-10 dataset.**

| Category | YOLOv8 AP (%) | LI-YOLOv8 AP (%) | Ap improvement rate (%) |
|---|---|---|---|
| airplane | 95.2 | 96.4 | 1.2 |
| ship | 89.0 | 91.9 | 2.9 |
| storage tank | 93.0 | 94.2 | 1.2 |
| baseball diamond | 96.5 | 97.3 | 0.8 |
| tennis court | 78.5 | 80.4 | 1.9 |
| basketball court | 68.4 | 79.6 | 11.2 |
| ground track field | 94.7 | 96.8 | 2.1 |
| harbor | 82.2 | 84.7 | 2.5 |
| bridge | 71.2 | 80.0 | 8.8 |
| vehicle | 73.9 | 68.7 | -5.2 |

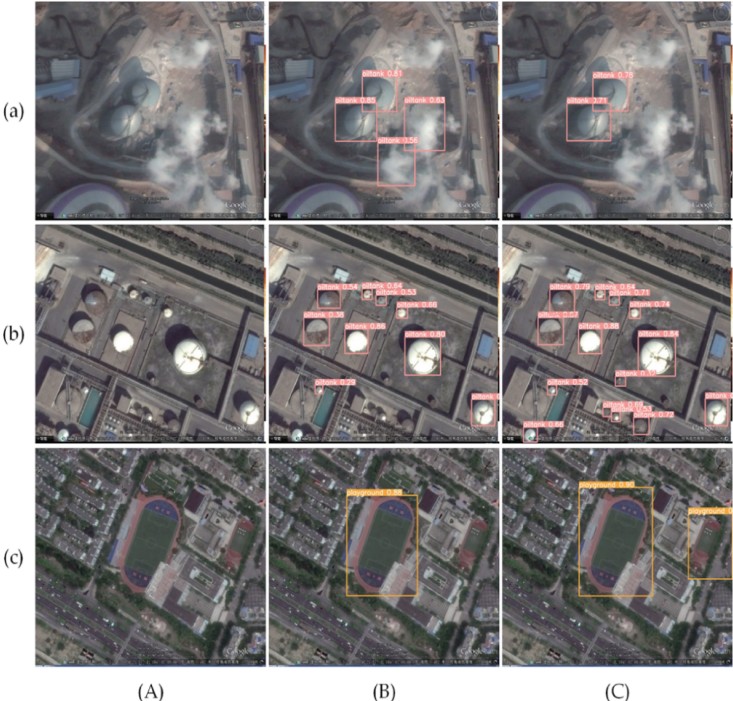

**Fig 13. Visualization comparison on RSOD dataset.** (a), (b), and (c) represent complex background targets, tiny targets, and multi-scale targets, whereas(A), (B), and (C) the original image, YOLOv8, and LI-YOLOv8, respectively.

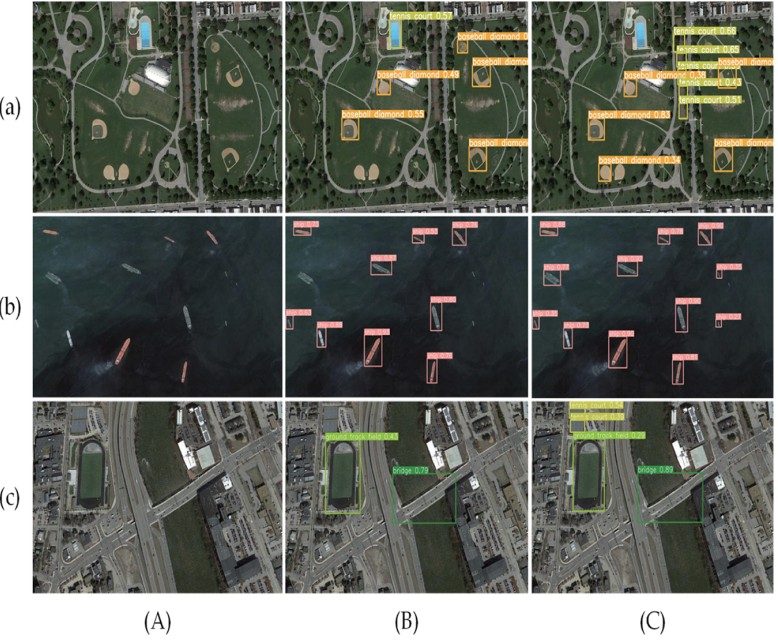

**Fig 14. Visualization comparison on NWPU VHR-10 dataset.** (a), (b), and (c) represent complex background targets, tiny targets, and multi-scale targets, whereas (A), (B), and (C) the original image, YOLOv8, and LI-YOLOv8.

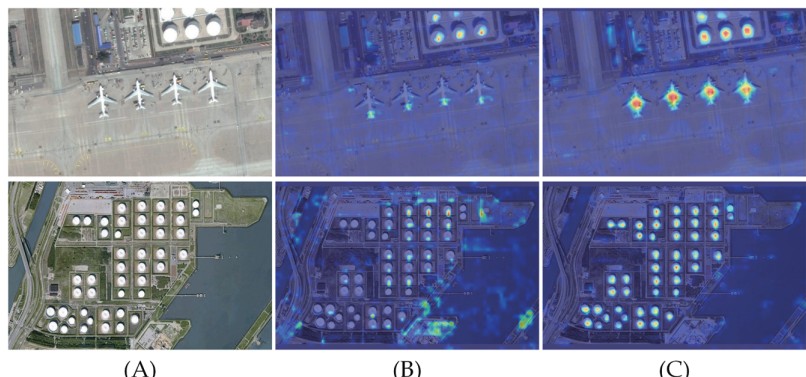

**Fig 15. Heatmap Comparison.** ((A), (B), and (C) the heatmap of YOLOv8, and the heatmap of LI-YOLOv8.

demonstrates a higher coverage rate of red areas for small targets, demonstrating a higher level of attention to these small targets.

(4) Numerical Comparison

1) We conducted comparative experiments using the RSOD dataset alongside YOLOv3-tiny, YOLOv5n, YOLOv5s, YOLOv6s, YOLOv8n, YOLOv9c [31], YOLOv10n [32] and LI-YOLOv8. The results obtained from experiments are showcased in Table 5, followed by the analysis.

YOLOv3-tiny exhibits mAP@0.5, mAP@0.5:0.95, and Precision values that are 3.1%, 2.7%, and 6.7% lower than those of LI-YOLOv8, respectively. It also possesses 9.4M more Parameters and 12.7G more GFLOPs, although its Recall is 2.0% higher. This suggests that, aside from Recall, YOLOv3-tiny underperforms in other performance metrics and demands higher hardware storage capacity.

YOLOv5n shows an increase in mAP@0.5:0.95 and Precision by 1.8% and 9.7% compared to YOLOv3-tiny, but it has lower values for mAP@0.5 and Recall by 4.4% and 18.3%, respectively. Its Parameters and GFLOPs are lower at 9.6M and 11.8G. Compared to LI-YOLOv8, YOLOv5n is lower in mAP@0.5, mAP@0.5:0.95, and Recall by 7.5%, 0.9%, and 16.3%, respectively, while Precision is 3.0% higher, with Parameters slightly lower by 0.2M and GFLOPs higher by 0.9G, suggesting that multiple metrics of YOLOv5n require further enhancement.

YOLOv5s performs better than YOLOv5n in mAP@0.5, mAP@0.5:0.95, and Recall, showing increases of 5.5%, 1.6%, and 13.7%, respectively. However, its Precision is 3.1% lower, with a substantial increase in Parameters to 21.5M and a slight rise in GFLOPs to 2.0G. Compared to LI-YOLOv8, YOLOv5s is lower in mAP@0.5, Precision, and Recall by 2.0%, 0.1%,

**Table 5. Algorithm comparison experiment on the RSOD dataset.**

| Method | mAP@0.5 (%) | mAP@0.5:0.95 (%) | Precision (%) | Recall (%) | Parameters (M) | GFLOPs (G) |
|---|---|---|---|---|---|---|
| YOLOv3-tiny | 88.8 | 64.3 | 88.4 | 88.0 | 12.1 | 19.0 |
| YOLOv5n | 84.4 | 66.1 | 98.1 | 69.7 | 2.5 | 7.2 |
| YOLOv5s | 89.9 | 67.7 | 95.0 | 83.4 | 24.0 | 9.2 |
| YOLOv6s | 89.7 | 67.5 | 91.9 | 84.6 | 16.3 | 44.2 |
| YOLOv8n | 84.3 | 64.9 | 90.6 | 69.4 | 3.0 | 8.2 |
| YOLOv9c | 91.7 | 69.3 | 94.8 | 86.0 | 25.5 | 103.0 |
| YOLOv10n | 90.1 | 65.0 | 93.0 | 84.1 | 2.7 | 8.4 |
| LI-YOLOv8(Ours) | 91.9 | 67.0 | 95.1 | 86.0 | 2.7 | 6.3 |

and 2.6%, respectively, while mAP@0.5:0.95 is higher by 0.7%. Its Parameters and GFLOPs are also greater by 21.3M and 2.9G, indicating that YOLOv5s is more complex and challenging to deploy.

YOLOv6s performs marginally below YOLOv5s in terms of mAP@0.5, mAP@0.5:0.95, and Precision, with reductions of 0.2%, 0.2%, and 3.1%, respectively. However, its Parameters and GFLOPs are excessively high at 16.3M and 44.2G. Compared to LI-YOLOv8, YOLOv6s is lower in mAP@0.5, Precision, and Recall by 2.2%, 3.2%, and 1.%, respectively, and mAP@0.5:0.95 is lower by 0.5%. Additionally, its Parameters and GFLOPs increase significantly by 13.6M and 37.9G, indicating a substantial computational burden and deployment challenges.

YOLOv8n shows lower values in mAP@0.5, mAP@0.5:0.95, Precision, and Recall than YOLOv6s by 4.8%, 2.6%, 1.3%, and 15.2%, respectively, with Parameters and GFLOPs that are 13.3M and 36G lower. Compared to LI-YOLOv8, YOLOv8n is lower in all metrics by 7.6%, 2.1%, 4.5%, and 16.6%, while its Parameters and GFLOPs are higher by 0.3M and 1.9G. This indicates that LI-YOLOv8 is smaller in scale and offers superior detection rates.

YOLOv9c achieves mAP@0.5, mAP@0.5:0.95, Precision, and Recall values of 91.7%, 69.3%, 94.8%, and 86.0%, respectively, which significantly surpass the first five algorithms. However, its mAP@0.5 and Precision are lower than the LI-YOLOv8 by 0.2% and 0.3%, while its mAP@0.5:0.95 is higher by 2.3%. It also has the highest Parameters and GFLOPs among the compared algorithms, reaching 25.5M and 103.0G, making it difficult to deploy.

Although YOLOv10n shows a significant improvement in Parameters and GFLOPs compared to YOLOv9c, being lower by 22.8M and 94.6G, its mAP@0.5, mAP@0.5:0.95, Precision, and Recall are lower by 1.6%, 4.3%, 1.8%, and 1.9%, respectively. Compared to LI-YOLOv8, it is lower in all metrics by 1.8%, 2.0%, 2.1%, and 1.9%, indicating that several metrics need improvement.

LI-YOLOv8 outperforms the seven algorithms mentioned above, achieving mAP@0.5, mAP@0.5:0.95, Precision, and Recall values of 91.9%, 67.0%, 95.1%, and 86.0%, respectively. It also significantly reduces the Parameters and GFLOPs to just 2.7M and 6.3G. The aggregate experimental results suggest that LI-YOLOv8 achieves superior performance in detecting small objects in remote sensing imagery.

2) On the NWPU VHR-10 dataset, comparative experiments are performed using YOLOv3-tiny, YOLOv5n, YOLOv8n, YOLOv10n, and LI-YOLOv8. The findings can be found in Table 6, followed by the analysis.

YOLOv3-tiny has a Recall that is 3.3% higher than the LI-YOLOv8, but it shows lower values in mAP@0.5, mAP@0.5:0.95, and Precision by 0.1%, 0.4%, and 4.1%, respectively. This indicates that, aside from Recall, other performance metrics of YOLOv3-tiny require improvement.

YOLOv5n's Precision is 2.2% superior to YOLOv3-tiny; nevertheless, it registers lower scores in mAP@0.5, mAP@0.5:0.95, and Recall by 3.3%, 1.0%, and 8.3%, respectively. When

**Table 6. Algorithm comparison experiment on the NWPU VHR-10 dataset.**

| Method | mAP@0.5 (%) | mAP@0.5:0.95 (%) | Precision (%) | Recall (%) |
|---|---|---|---|---|
| YOLOv3-tiny | 87.0 | 54.5 | 91.7 | 80.2 |
| YOLOv5n | 84.1 | 53.5 | 93.9 | 71.9 |
| YOLOv8n | 84.3 | 53.8 | 92.0 | 74.6 |
| YOLOv10n | 78.7 | 46.7 | 72.5 | 71.6 |
| LI-YOLOv8(Ours) | 87.1 | 54.9 | 95.8 | 76.9 |

compared to our method, YOLOv5n displays inferior metrics across all indicators by 3.0%, 1.9%, 1.9%, and 5.0%, demonstrating its weaker detection capabilities.

YOLOv8n's metrics are close to those of YOLOv5n, with mAP@0.5, mAP@0.5:0.95, and Recall only 0.2%, 0.3%, and 3.0% higher, respectively, while Precision is 1.9% lower. However, all metrics of YOLOv8n are still below those of the LI-YOLOv8, lower by 2.8%, 1.1%, 3.8%, and 2.3%.

YOLOv10n's metrics are lower than those of LI-YOLOv8, achieving only 78.7%, 46.7%, 72.%, and 71.6% for mAP@0.5, mAP@0.5:0.95, Precision, and Recall, respectively. These values are lower than the LI-YOLOV8 by 8.4%, 8.2%, 23.3%, and 5.3%, indicating a low detection rate for YOLOv10n.

Compared to the aforementioned algorithms, LI-YOLOv8 outperforms the others with mAP@0.5, mAP@0.5:0.95, and Precision values of 87.1%, 54.9%, and 95.8%, respectively. Additionally, as shown in Table 5, the LI-YOLOv8 has the lowest number of parameters and computational load.

Considering the analysis presented, the average precision of LI-YOLOv8 has significantly improved, with a reduction in the number of parameters and computational cost, and it exhibits a stronger recognition capability for small targets in remote sensing imagery.

**Generalization experiment.** Generalization experiments were conducted using YOLOv8 and LI-YOLOv8 on the TinyPerson, LEVIR-ship, brain-tumor, and smoke_fire_1 datasets. The results are presented in Table 7, and the analysis is discussed below.

On the TinyPerson dataset, LI-YOLOv8 achieved improvements of 2.6% in mAP@0.5, 0.2% in mAP@0.5:0.95, and 7.0% in Precision, compared to YOLOv8. On the LEVIR-ship dataset, LI-YOLOv8 outperformed YOLOv8 with increases of 5.3% in mAP@0.5, 3.3% in mAP@0.5:0.95, and 0.1% in Precision. These results demonstrate that LI-YOLOv8 is not only effective for the RSOD and NWPU VHR-10 datasets but also enhances the detection of small targets in complex scenarios across other remote sensing datasets.In the brain-tumor dataset, generalization experiments comparing YOLOv8 and LI-YOLOv8 revealed that LI-YOLOv8 improved mAP@0.5 by 2.6%, mAP@0.5:0.95 by 3.5%, and Precision by 3.7%. This indicates that LI-YOLOv8 is effective not only for specific target detection tasks in remote sensing images but also enhances the detection of non-small targets in medical images. Similarly, on the smoke_fire_1 dataset, LI-YOLOv8 achieved increases of 2.3% in mAP@0.5, 1.3% in mAP@0.5:0.95, and 4.4% in Precision compared to YOLOv8. These improvements validate that the LI-YOLOv8 algorithm effectively detects fire and smoke targets in urban surveillance settings, significantly enhancing their detection rates.In summary, LI-YOLOv8 not only demonstrates superior performance in detecting targets within remote sensing images but also exhibits outstanding results in detecting medium and large-sized targets across various other domains, highlighting its versatility and general applicability.

**Table 7. Generalization experiment on the TinyPerson dataset.**

| Dataset | Method | mAP@0.5 (%) | mAP@0.5:0.95 (%) | Precision (%) |
|---|---|---|---|---|
| TinyPerson | YOLOv8 | 44.2 | 21.8 | 85.8 |
| | LI-YOLOv8 | 46.8 | 22.0 | 92.8 |
| LEVIR-ship | YOLOv8 | 72.0 | 29.8 | 79.6 |
| | LI-YOLOv8 | 77.3 | 33.1 | 79.7 |
| brain-tumor | YOLOv8 | 42.8 | 31.9 | 50.9 |
| | LI-YOLOv8 | 45.4 | 35.4 | 54.6 |
| smoke_fire_1 | YOLOv8 | 80.8 | 58.0 | 70.1 |
| | LI-YOLOv8 | 83.1 | 59.3 | 74.5 |

## Conclusions

To address the challenges in identifying small targets in remote sensing images, such as difficulties in feature extraction, confusion between background and targets, significant deviations in prediction boxes, high rates of missed detections and false positives, computational complexity, and high resource consumption, we propose a lightweight small target detection algorithm for remote sensing images that combines GSConv and PConv, named LI-YOLOv8. We enhance feature extraction by improving SPPF to SPPF-R, increase the focus on small target areas by upgrading the C2f in the neck network to C2f-E, design a lightweight detection head (GP-Detect) to reduce network complexity, and replace the bounding box loss function from CIoU to Inner-Wise IoU to improve the algorithm's generalization capability. Experimental results demonstrate that our proposed algorithm outperforms baseline methods and other recent YOLO algorithms in small target detection within remote sensing images, effectively enhancing detection performance and exhibiting strong generalizability.

Future research will continue to focus on optimizing model training time by reducing training duration while maintaining model lightweightness and accuracy. Additionally, efforts will be made to extend the model's applicability to a broader range of real-world scenarios, thereby enhancing its practical value and efficiency.

## Author contributions

**Conceptualization:** Pingping Yan.

**Data curation:** Pingping Yan.

**Formal analysis:** Pingping Yan.

**Investigation:** Liang Jiang.

**Methodology:** Pingping Yan.

**Resources:** Xiangming Qi, Liang Jiang.

**Software:** Pingping Yan, Liang Jiang.

**Supervision:** Xiangming Qi, Liang Jiang.

**Validation:** Pingping Yan.

**Visualization:** Pingping Yan, Liang Jiang.

**Writing – original draft:** Pingping Yan.

**Writing – review & editing:** Xiangming Qi.

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
