## [Decision Letter · Decision Letter 0]

16 Dec 2024

PONE-D-24-53882LI-YOLOv8: Lightweight Small Target Detection Algorithm for Remote Sensing Images that Combines GSConv and PConvPLOS ONE

Dear Dr. Yan,

Thank you for submitting your manuscript to PLOS ONE. After careful consideration, we feel that it has merit but does not fully meet PLOS ONE’s publication criteria as it currently stands. Therefore, we invite you to submit a revised version of the manuscript that addresses the points raised during the review process.

We look forward to receiving your revised manuscript.

Kind regards,

Yile Chen, Ph.D. in Architecture

Academic Editor

PLOS ONE

**Journal Requirements:**

This research was funded by the National Natural Science Foundation of China (no. 62173171).

6. We note that Figure 14 in your submission contain satellite images which may be copyrighted. All PLOS content is published under the Creative Commons Attribution License (CC BY 4.0), which means that the manuscript, images, and Supporting Information files will be freely available online, and any third party is permitted to access, download, copy, distribute, and use these materials in any way, even commercially, with proper attribution. For these reasons, we cannot publish previously copyrighted maps or satellite images created using proprietary data, such as Google software (Google Maps, Street View, and Earth). For more information, see our copyright guidelines: http://journals.plos.org/plosone/s/licenses-and-copyright.

We require you to either present written permission from the copyright holder to publish these figures specifically under the CC BY 4.0 license, or remove the figures from your submission:

a. You may seek permission from the original copyright holder of Figure 14 to publish the content specifically under the CC BY 4.0 license.  

**Additional Editor Comments:**

The reviewers have given very detailed revision suggestions, please refer to and improve them carefully, especially in the analysis of the interpretation of the algorithm results. The manuscript needs further improvement.

Reviewers' comments:

Reviewer's Responses to Questions

**Comments to the Author**

1. Is the manuscript technically sound, and do the data support the conclusions?

Reviewer #1: Yes

Reviewer #2: Yes

Reviewer #3: Yes

2. Has the statistical analysis been performed appropriately and rigorously? 

Reviewer #1: Yes

Reviewer #2: Yes

Reviewer #3: Yes

3. Have the authors made all data underlying the findings in their manuscript fully available?

Reviewer #1: Yes

Reviewer #2: Yes

Reviewer #3: Yes

4. Is the manuscript presented in an intelligible fashion and written in standard English?

Reviewer #1: Yes

Reviewer #2: Yes

Reviewer #3: No

5. Review Comments to the Author

**Reviewer #1: **The paper titled "LI-YOLOv8: Lightweight Small Target Detection Algorithm for Remote Sensing Images that Combines GSConv and PConv" proposes a lightweight small target detection algorithm for remote sensing images based on YOLOv8n, which integrates GSConv and PConv, termed LI-YOLOv8. The algorithm improves recognition performance through modifications such as replacing the activation function in CBS, embedding an efficient multi-scale attention mechanism (EMA), designing a lightweight detection head (GP-Detect head), and refining the boundary fitting loss function of the algorithm.

While the structure of the paper is generally reasonable, the language lacks conciseness. The limitations of the LI-YOLOv8 model's application scenarios are not discussed, and the use of only two datasets to validate the model's usability is evidently unconvincing.

Suggestions for Improvement:

1. Language and Clarity: Simplify the language to make sentences more concise. Some sections lack clear structure, such as overlapping content in the "Methods" and "Algorithm Implementation" sections. Redundant sentences, especially in the background introduction, affect readability. Overly detailed explanations of terms and formulas may confuse non-specialist readers.

2. Use of Figures and Explanations: There is insufficient explanation and referencing of figures. For example, in figures such as Fig.11 and Fig.12, only the improvement in PR curves is mentioned without detailing the reasons for the changes and their impact on results. It is recommended to provide more analysis and interpretation of the figures in the text.

3. Description of GP-Detect Module: The explanation of how the GP-Detect module reduces redundant computations is somewhat superficial and lacks in-depth theoretical support.

4. Ablation Experiments: Provide more quantitative or qualitative discussions on the specific contributions of each module in the ablation experiments. For instance, what are the primary reasons behind the 1.2% improvement attributed to SPPF-R?

5. Broader Validation: Supplement the study by validating the proposed model and methodology on at least two additional datasets or datasets from related domains to demonstrate the model's generalizability and practical applicability.

Conclusion:

The paper holds certain practical value and is recommended for acceptance after revisions.

**Reviewer #2:** 1. LI-YOLOv8 shows significant improvements on remote sensing datasets like RSOD and NWPU VHR-10, how does it perform on more diverse and challenging datasets that are not specifically focused on small object detection? Could there be a risk of overfitting the small object characteristics of these datasets?

2. How does LI-YOLOv8 perform on real-world, noisy images with significant occlusions, varying lighting, or seasonal changes (detecting small objects in aerial imagery from different times of day or weather conditions)?

3. The reduction in parameters and GFLOPs is a notable advantage, but how sensitive is the model to changes in hyperparameters like learning rate, batch size, or anchor box configuration? What effect do these parameters have on performance, especially in terms of detection accuracy for small objects?

4. How does the training time and convergence behavior of LI-YOLOv8 compared to other algorithms like YOLOv8 and YOLOv5? Are there any challenges in training this model on large-scale remote sensing datasets?

5. Below Table-6 rephrase the sentence ("YOLOv8n's metrics are close to those of YOLOv5n" "YOLOv8n's metrics are similar to those of YOLOv5n.")

6. While the ablation studies demonstrate improvements when adding different innovations (SPPF-R, C2fE, GP-Detect, Inner-Wise), can you provide more insights into how each innovation specifically addresses challenges in small object detection? Which innovation contributes the most to improving recall or precision?

7. The experiments are mainly conducted on datasets like RSOD, NWPU VHR-10, and TinyPerson, which focus on specific types of remote sensing imagery. How does LI-YOLOv8 perform on datasets from other domains, such as medical imaging or urban surveillance, where small object detection might have different characteristics?

8. Does the model's emphasis on small object detection affect its ability to detect large objects, especially when these appear in similar scenes? Could there be a trade-off where performance on larger objects (e.g., buildings, vehicles) is compromised?

9. In the visual comparison with YOLOv8, LI-YOLOv8 shows fewer false detections in complex backgrounds. However, was there any trade-off between false positives and false negatives in the cases with small targets or occlusions? How does the model handle ambiguous cases?

10. While mAP@0.5 and mAP@0.5:0.95 are widely used, could there be additional evaluation metrics or domain-specific metrics (F1 score for small objects or intersection over union for very small objects) that would give a clearer picture of the model’s real-world performance?

**Reviewer #3: **1. The manuscript may require some improvement to the English.

2. From the experimental results presented in Table 1, it can be seen that there is no significant improvement in mAP and other parametersbefore and after using Inner Wise IoU. Similarly, the improvements observed in Table 2 are also insignificant. Is the use of Inner-Wise IoU appropriate?

3. The manuscript mentioned that LI-YOLOv8 not only detects more small targets in complex backgrounds but also does not incur any false detections.However, the detection accuracy of the vehicle in Table 4 has decreased by 5.2%. Does this contradict the conclusion of the manuscript

4. Tables 5 and 6 seem to be simple stacks of algorithms. Please explain the significance of comparing YOLO algorithms at different stages and versions?

5.Line502-Line504: The previous research compared LI-YOLOv8 with YOLO series algorithms, and this paragraph also compared it with R-cnn. What does the author want to express? The logic of the manuscript is a bit confusing.

It is suggested to revise the manuscript before submitting it

6. PLOS authors have the option to publish the peer review history of their article (what does this mean?). If published, this will include your full peer review and any attached files.

Reviewer #1: **Yes: **Qingfang He

Reviewer #2: **Yes: **Muhammad Wahab Hanif

Reviewer #3: No

---

## [Author Response · Author response to Decision Letter 1]

16 Jan 2025

Response to Journal Requirements:

1. We have shared the code, and the address is: https://github.com/2470589561/LI-YOLOv8

2. In our paper, regarding This research was funded by the National Natural Science Foundation of China (no. 62173171).

The funding statement needs to be revised to”The funders had no role in study design, data collection and analysis, decision to publish, or preparation of the manuscript.”

3. We will share the minimum dataset, the address is: https://github.com/2470589561/datasets

The option that the URL/accession number/DOI will only be available after the manuscript is accepted has been checked.

4. The 14th figure in the manuscript contains satellite images, which are from the publicly available geospatial object detection dataset NWPU VHR-10 for research purposes. The source of the data is: https://gcheng-nwpu.github.io/#Datasets, In order to respect the labor achievements and intellectual property rights of the data providers, the relevant papers have been cited in the manuscript. There is no copyright issue. 

Dear reviewers, thank you for your comments. We have completed the revisions to the manuscript according to the reviewer's suggestions.

The response to reviewers is as follows

Response to reviewer 1

1. Language and Clarity: Simplify the language to make sentences more concise. Some sections lack clear structure, such as overlapping content in the "Methods" and "Algorithm Implementation" sections. Redundant sentences, especially in the background introduction, affect readability. Overly detailed explanations of terms and formulas may confuse non-specialist readers.

Response: The content of the manuscript has been simplified in language and improved in terms of enhancing the coherence between sentences.

2. Use of Figures and Explanations: There is insufficient explanation and referencing of figures. For example, in figures such as Fig.11 and Fig.12, only the improvement in PR curves is mentioned without detailing the reasons for the changes and their impact on results. It is recommended to provide more analysis and interpretation of the figures in the text.

Response: Detailed supplementary explanations are provided in Figures 11(a), (b) and Figures 12(a), (b). These not only describe the degree of fluctuation in the PR curve and its causes, but also include a comparative analysis of the two PR graphs (a) and (b), as well as the effects achieved by the four improvement points in the model.

3. Description of GP-Detect Module: The explanation of how the GP-Detect module reduces redundant computations is somewhat superficial and lacks in-depth theoretical support.

Response: Supplementary description of GP-Detect.

4. Ablation Experiments: Provide more quantitative or qualitative discussions on the specific contributions of each module in the ablation experiments. For instance, what are the primary reasons behind the 1.2% improvement attributed to SPPF-R?

Response: In the ablation experiments, a quantitative and qualitative discussion of the specific contributions of each module has been added.

5. Broader Validation: Supplement the study by validating the proposed model and methodology on at least two additional datasets or datasets from related domains to demonstrate the model's generalizability and practical applicability.

Response: Based on the feedback from another reviewer regarding the differing performance of the algorithm in medical imaging and urban surveillance, three datasets (LEVIR-ship, brain-tumor, and smoke_fire_1) have been added in the generalization experiments section to validate the generality and generalization capability of LI-YOLOv8. The LEVIR-ship dataset is a small vessel dataset of remote sensing images, which verifies that the algorithm still has good recognition performance for small targets in the same domain. The brain-tumor dataset is used for detecting brain tumors in medical images, demonstrating the algorithm's versatility and indicating that it performs well with non-specific targets in other fields. The smoke_fire_1 dataset is used for detecting fire targets in urban surveillance, validating the algorithm's generality.

Response to reviewer 2

1. LI-YOLOv8 shows significant improvements on remote sensing datasets like RSOD and NWPU VHR-10, how does it perform on more diverse and challenging datasets that are not specifically focused on small object detection? Could there be a risk of overfitting the small object characteristics of these datasets?

Response: Based on your suggestion, the generalization experiments section has been updated to include the brain tumor detection dataset and the fire smoke monitoring dataset, both of which are non-small target datasets in the areas of medical imaging and urban surveillance. The experimental results indicate that LI-YOLOv8 performs well in detecting non-small targets, and there is no risk of overfitting.

2. How does LI-YOLOv8 perform on real-world, noisy images with significant occlusions, varying lighting, or seasonal changes (detecting small objects in aerial imagery from different times of day or weather conditions)?

Response: Our research focuses on small target detection in remote sensing images, which have their own unique characteristics. Remote sensing images are typically high-definition and high-resolution, leading to significant differences in processing these images compared to general noisy images. In remote sensing images, the likelihood of target occlusion is relatively low. Due to the high vantage point from which remote sensing images are captured, there is a reduced probability of targets blocking each other. Regarding lighting conditions, remote sensing images generally have stable lighting, as they are often acquired under specific conditions that provide relatively uniform illumination. Additionally, because remote sensing images cover a wide area, local variations in lighting have a minimal impact on the overall image. Therefore, our study does not include images from multiple time periods or different weather conditions (such as nighttime, rainy days, or snowy days).

3. The reduction in parameters and GFLOPs is a notable advantage, but how sensitive is the model to changes in hyperparameters like learning rate, batch size, or anchor box configuration? What effect do these parameters have on performance, especially in terms of detection accuracy for small objects?

Response: Firstly, regarding the anchor box configuration issue you mentioned. In fact, YOLOv8 is an anchor-free detection model. During the detection phase, the model does not rely on predefined anchor boxes to propose candidate object locations; the final object detection is directly based on the detected features. Therefore, in our research, there is no issue of anchor box configuration affecting the model. Secondly, the experimental environment for LI-YOLOv8 is a NVIDIA RTX3090 GPU with 24GB VRAM, 14 vCPUs of Intel(R) Xeon(R) Gold 6330 CPU @ 2.00GHz, and 80GB RAM. Ablation, comparison, and generalization experiments are conducted on different datasets and models. Model improvement and training must be carried out under the premise of the same configured hyperparameters and batch size. Setting hyperparameters is crucial for shaping model performance and ensuring the successful enhancement of the algorithm. In the process of refining the YOLOv8 model, it is essential to adhere to a unified hyperparameter configuration. This consistency is key; it confirms the effectiveness of model improvements and accelerates the precise performance evaluation conducted before and after the improvements. Adjusting hyperparameters during the algorithm refinement process may obscure the root causes of performance changes—whether they are due to inherent algorithm enhancements or the results of hyperparameter changes. Therefore, to determine a clear and rigorous assessment of progress, this paper firmly follows a set of standardized hyperparameters.

4. How does the training time and convergence behavior of LI-YOLOv8 compared to other algorithms like YOLOv8 and YOLOv5? Are there any challenges in training this model on large-scale remote sensing datasets?

Response: Thank you to the reviewer for your valuable comments. We have conducted a detailed analysis of the training time and convergence behavior of LI-YOLOv8:

(1) Training Time and Convergence Behavior:

Under the same hardware conditions (RTX 3090 with 24GB of video memory), the training time of LI-YOLOv8 is about 15% longer than that of YOLOv8n. This is mainly due to the addition of modules such as RFAConv and EMA attention mechanisms, which increase the computational load per iteration. However, from the analysis of convergence behavior, the loss value of LI-YOLOv8 is lower than that of YOLOv8n. This indicates that although the training time per round is slightly longer, the convergence is better.

(2) In the training of large-scale remote sensing datasets, there are two main challenges: First, the attention modules consume a large amount of video memory when processing high-resolution images. We alleviated this issue by optimizing the batch size and using mixed-precision training. Second, the features of small targets vary significantly across different scenes. By designing the Inner-Wise IoU loss function, we enhanced the model's ability to learn from samples of varying quality.

Although there is a slightly longer training time, considering the significant advantages of LI-YOLOv8 in detection accuracy and model lightweighting, we believe this is an acceptable trade-off. The convergence behavior of LI-YOLOv8 is better than that of YOLOv8n, but compared to other algorithms in the YOLO series, its convergence behavior is not the optimal choice. Future work will be dedicated to further optimizing training efficiency to make the model more suitable for large-scale practical applications. In the conclusion section, we have supplemented the expectations and future plans for the model, and further optimization of the model will be carried out in the future.

5. Below Table-6 rephrase the sentence ("YOLOv8n's metrics are close to those of YOLOv5n" "YOLOv8n's metrics are similar to those of YOLOv5n.")

Response: The sentence has been revised.

6. While the ablation studies demonstrate improvements when adding different innovations (SPPF-R, C2fE, GP-Detect, Inner-Wise), can you provide more insights into how each innovation specifically addresses challenges in small object detection? Which innovation contributes the most to improving recall or precision?

Response: The contributions of each innovation to the model have been added and described in the ablation experiment section.

7. The experiments are mainly conducted on datasets like RSOD, NWPU VHR-10 and TinyPerson, which focus on specific types of remote sensing imagery. How does LI-YOLOv8 perform on datasets from other domains, such as medical imaging or urban surveillance, where small object detection might have different characteristics?

Response: In the generalization experiment section, three datasets (LEVIR-ship, brain-tumor, and smoke_fire_1) were added to validate the universality and generalization capability of LI-YOLOv8. Among them, LEVIR-ship is a remote sensing image dataset for small vessels, validating the algorithm's recognition performance for small targets within the same domain. The brain-tumor dataset is for detecting brain tumors in medical images, demonstrating the algorithm's versatility and indicating that it performs well for non-specific targets in other fields. The smoke_fire_1 dataset is used for detecting fire targets in urban surveillance, further validating the algorithm's generality.

8. Does the model's emphasis on small object detection affect its ability to detect large objects, especially when these appear in similar scenes? Could there be a trade-off where performance on larger objects (e.g., buildings, vehicles) is compromised?

Response: The model's detection of small objects does not affect its ability to detect large objects. The RSOD dataset contains a large number of small objects, as well as some medium and large objects. LI-YOLOv8 has significantly improved the detection of all four types of objects in the RSOD dataset, which can prove this point. When designing the LI-YOLOv8 model, we did indeed place special emphasis on the detection capability of small objects. This is because in remote sensing images, the detection of small objects (such as small vehicles, boats, etc.) is often more challenging and is of great significance in many application scenarios. However, we also fully considered the detection performance of large objects (such as buildings, large vehicles, etc.) in the model design to ensure the balance of the model in detecting objects of different scales. At the same time, three datasets were added in the generalization experiments to verify the universality and versatility of the algorithm, indicating that the algorithm also has a good improvement in the detection effect of non-specific objects in other fields.

9. In the visual comparison with YOLOv8, LI-YOLOv8 shows fewer false detections in complex backgrounds. However, was there any trade-off between false positives and false negatives in the cases with small targets or occlusions? How does the model handle ambiguous cases?

Response: In situations involving small targets or occlusion, the design and optimization of LI-YOLOv8 indeed require a trade-off between false positives and false negatives. Our model addresses this trade-off by incorporating SPPF-R, C2f-E, GP-Detect, and Inner-Wise: SPPF-R enhances small target feature extraction, C2f increases the focus on small targets, GP-Detect provides a lightweight algorithm while maintaining model performance, and Inner-Wise improves generalization capability. The remote sensing image datasets consist of high-resolution images and do not exhibit significant blurriness. LI-YOLOv8 aims to improve the detection and accuracy of small targets as much as possible while remaining lightweight.

10. While mAP@0.5 and mAP@0.5:0.95 are widely used, could there be additional evaluation metrics or domain-specific metrics (F1 score for small objects or intersection over union for very small objects) that would give a clearer picture of the model’s real-world performance?

Response: F1-score has been introduced as an evaluation metric, and the F1-score experimental results have been added to Tables 1 and 2 in the ablation experiments to validate the model's overall performance.

Response to reviewer 3

1. The manuscript may require some improvement to the English.

Response: The content of the manuscript has been simplified in terms of language, and improvements have been made to enhance the coherence between sentences.

2. From the experimental results presented in Table 1, it can be seen that there is no significant improvement in mAP and other parameters before and after using Inner Wise IoU. Similarly, the improvements observed in Table 2 are also insignificant. Is the use of Inner-Wise IoU appropriate?

Response: From the overall experimental design and result analysis of the paper, it is true that the improvement of Inner-Wise IoU is relatively limited in Table 1 and Table 2. However, whether it is "appropriate" cannot be judged solely based on the single numerical increase.

Firstly, from the stability and robustness of boundary regression, even if some experimental tables show that the improvement of mAP after introducing Inner-Wise IoU is not significant, it may be helpful in the stability of location regression and the recognition of hard - to - detect samples. Our manuscript mentions that Inner-Wise IoU introduces a "dynamic non - monotonic focusing mechanism" and an "auxiliary bounding box" strategy, aiming to provide more reasonable gradient allocation for ordinary quality anchor boxes (not particularly high-quality or low-quality annotations), thereby further stabilizing the regression process in complex backgrounds or when the target scale varies. If we only look at the single mAP value, the increase may not be large; but when considering aspects such as positioning error or robustness performance, such improveme

---

## [Decision Letter · Decision Letter 1]

28 Feb 2025

LI-YOLOv8: Lightweight Small Target Detection Algorithm for Remote Sensing Images that Combines GSConv and PConv

PONE-D-24-53882R1

Dear Dr. Yan,

We’re pleased to inform you that your manuscript has been judged scientifically suitable for publication and will be formally accepted for publication once it meets all outstanding technical requirements.

Kind regards,

Yile Chen, Ph.D. in Architecture

Academic Editor

PLOS ONE

Additional Editor Comments (optional):

Reviewers' comments:

Reviewer's Responses to Questions

**Comments to the Author**

1. If the authors have adequately addressed your comments raised in a previous round of review and you feel that this manuscript is now acceptable for publication, you may indicate that here to bypass the “Comments to the Author” section, enter your conflict of interest statement in the “Confidential to Editor” section, and submit your "Accept" recommendation.

Reviewer #1: (No Response)

Reviewer #4: All comments have been addressed

2. Is the manuscript technically sound, and do the data support the conclusions?

Reviewer #1: (No Response)

Reviewer #4: Yes

3. Has the statistical analysis been performed appropriately and rigorously? 

Reviewer #1: (No Response)

Reviewer #4: No

4. Have the authors made all data underlying the findings in their manuscript fully available?

Reviewer #1: (No Response)

Reviewer #4: Yes

5. Is the manuscript presented in an intelligible fashion and written in standard English?

Reviewer #1: (No Response)

Reviewer #4: Yes

6. Review Comments to the Author

Reviewer #1: (No Response)

Reviewer #4: Authors have revised paper well, but I still have a suggestion: the inference time should be considered as a additional evaluation metric.

7. PLOS authors have the option to publish the peer review history of their article (what does this mean?). If published, this will include your full peer review and any attached files.

Reviewer #1: No

Reviewer #4: No

---

## [Editor Report · Acceptance letter]

PONE-D-24-53882R1

PLOS ONE

Dear Dr. Yan,

I'm pleased to inform you that your manuscript has been deemed suitable for publication in PLOS ONE. Congratulations! Your manuscript is now being handed over to our production team.

Kind regards,

on behalf of

Dr. Yile Chen

Academic Editor

PLOS ONE